# Socio-demographic and environmental determinants of camel mortality in Somaliland's nomadic communities: A negative binomial regression analysis of the 2020 demographic and health survey

Mohamed Ahmed Hassan[1], Omran Salih[1,2]*

**1** School of Postgraduate Studies & Research, Amoud University, Borama, Somalia, **2** Institute of Systems Science, Durban University of Technology, Durban, South Africa

\* omran.salih@amoud.edu.so

## Abstract

This study presents the first comprehensive quantitative assessment of the socio-demographic and environmental determinants of camel mortality in Somaliland's nomadic communities, leveraging household-level data from the 2020 Somaliland Demographic and Health Survey (SLDHS). Employing a Negative Binomial Regression analysis, we explored the magnitude of total and cause-specific (drought, flood, disease) camel mortality. Our findings reveal that camel mortality is not a random occurrence but is systematically influenced by a confluence of environmental and demographic factors. Key determinants of increased mortality risk include residence in the eastern regions of Somaliland (Togdheer, Sool, and Sanaag), older age of the household head, and reliance on unimproved water sources. Environmental shocks, particularly drought, emerged as the most significant drivers of camel deaths. While older household heads and larger household sizes were associated with fewer drought-related deaths, possibly due to accumulated experience or better coping strategies, female headship showed a nuanced role, being protective for overall and drought mortality but a risk factor for flood and disease-related deaths. Access to unimproved water sources consistently increased the risk of camel mortality across all causes, while higher household wealth was associated with significantly fewer deaths. The study highlights a clear and statistically significant east-west disparity in camel survival, with eastern regions experiencing higher mortality. These insights provide crucial evidence for policymakers and animal health organizations to design targeted interventions, safeguard pastoral assets, and strengthen community resilience against mounting environmental and economic pressures in data-scarce regions.

**Data availability statement:** The data used in this study are publicly available from the Somali National Bureau of Statistics through the Somali Health and Demographic Survey 2020. The dataset can be accessed at: https://microdata.nbs.gov.so/index.php/catalog/50.

**Funding:** The author(s) received no specific funding for this work.

**Competing interests:** The authors have declared that no competing interests exist.

## Introduction

The dromedary camel (Camelus dromedarius) represents a cornerstone of livelihood, food security, and cultural identity for millions of people residing in arid and semi-arid lands (ASAL), particularly within pastoralist communities. Uniquely adapted to withstand harsh environmental conditions such as extreme heat, water scarcity, and poor-quality forage, camels are vital assets where other livestock species often struggle to survive [1,2]. For these communities, camels are a multifaceted resource, providing essential sustenance through milk and meat, transport across difficult terrains, and a crucial source of income. Beyond their economic utility, they function as a primary form of wealth storage and hold significant social and cultural capital, playing a central role in traditional ceremonies, social status, and community resilience [3,4].

Despite their remarkable resilience, camel populations face significant production constraints that threaten herd sustainability and pastoral livelihoods. High mortality rates, especially among vulnerable young stock, are a persistent challenge, driven by a complex interplay of factors [5,6]. In the Somali region of Ethiopia, consistent with Somaliland, livestock keepers identified Camelpox as the top disease cause of camel calf mortality with a mean score of 4.75, followed by Calf Scour at 4.08. Contagious ecthyma, ticks, and pneumonic diseases ranked as the third, fourth, and fifth leading causes of mortality, respectively [3]. Furthermore, environmental stressors, intensified by climate change, including severe droughts and devastating floods, compound these risks, leading to widespread losses that undermine the very foundation of pastoral economies [1,7].

Globally, the dromedary camel population is estimated at around 35 million, with a supermajority over 80% concentrated in the arid and semi-arid regions of Africa [2,8]. The Horn of Africa, which includes Somalia, Ethiopia, Sudan, and Kenya, stands out as the world's primary camel hub, hosting approximately 60% of the continent's total camel population [1,5]. This highlights the species' profound regional significance. In recent years, camel populations have seen steady growth in East Africa, partly as a strategic adaptation by pastoralists responding to increased climate variability and rangeland degradation, which have negatively impacted more vulnerable livestock like camel [1,2].

Within this region, Somaliland's economy and the livelihoods of its predominantly nomadic and semi-nomadic populations are intrinsically linked to camel pastoralism. The traditional production system is characterized by mobility, following biannual rainy seasons in search of pasture and water [3]. However, camel production in Somaliland faces severe constraints, including limited access to formal veterinary services, reliance on traditional disease management, and extreme vulnerability to environmental shocks [2,4,9]. Recurrent and intensifying droughts lead to acute shortages of feed and water, weakening animals and increasing their susceptibility to disease, while seasonal floods can cause direct mortality and displace herds [1,2]. These challenges, coupled with high calf mortality rates reported to be between 10–45% in similar East African settings, severely limit herd growth and productivity [2,4,6].

While factors such as disease, drought, and poor husbandry are broadly acknowledged as drivers of camel mortality across East Africa, their specific magnitude and the relative importance of various socio-demographic and

environmental determinants within the unique context of Somaliland's nomadic communities remain poorly quantified. Existing research in the region often focuses on specific diseases or offers qualitative descriptions, lacking a comprehensive, data-driven analysis of the multifaceted drivers of mortality using robust statistical methods [8,9]. Crucially, there is a significant gap in understanding how household demographic characteristics, socioeconomic status, and environmental factors interact to influence camel mortality risk at the household level. Furthermore, the quantitative evidence of this mortality risk across Somaliland is largely unknown [9]. This lack of quantitative evidence hinders the development and effective targeting of veterinary interventions, resource allocation, and policies aimed at enhancing the resilience of pastoral livelihoods.

Therefore, this study aims to address these critical gaps by uniquely leveraging household-level data from the Somaliland Demographic and Health Survey (SLDHS) 2020. The primary objectives are: 1) to estimate the magnitude of total and cause-specific (drought, flood, disease) camel mortality among nomadic households in Somaliland; and 2) to identify the key socio-demographic and environmental determinants associated with the number of camel deaths using count regression models. The findings are intended to provide evidence-based insights for policymakers and animal health organizations to design targeted interventions that safeguard pastoral assets and strengthen community resilience against environmental and economic shocks [10].

## Materials and methods

### Study area

The study was conducted in Somaliland, a region in the Horn of Africa where nomadic and semi-nomadic pastoralism forms the backbone of the economy and livelihoods. Somaliland is part of the broader Horn of Africa, which hosts approximately 60% of Africa's camel population, making it a critical hub for camel pastoralism. Somaliland covers an estimated land area of 176,119.2km$^2$ and has a population of about 4.2 million people. Camels are vital assets in this arid and semi-arid environment, providing milk, meat, income, and transport, and serving as a key adaptation strategy to climate variability. However, the region is highly vulnerable to recurrent droughts, floods, and disease outbreaks, which significantly threaten camel survival and pastoral resilience. Understanding the patterns and determinants of camel mortality in this setting is essential to inform targeted interventions and strengthen the sustainability of pastoral livelihoods.

### Study design, data source, and population

This study employed a quantitative, cross-sectional design to analyze secondary data from the 2020 Somaliland Demographic and Health Survey (SLDHS) [https://microdata.nbs.gov.so/index.php/catalog/50]. The SLDHS is a nationally representative survey conducted by the Central Statistics Department and the Ministry of Health Development that provides high-quality data on a wide range of demographic, socioeconomic, and health-related indicators [11]. The cross-sectional design is analytical, seeking to identify the determinants associated with the reported counts of camel mortality at a specific point in time. The target population for this research comprised all nomadic and semi-nomadic households in Somaliland that reported owning camels during the survey reference period.

### Sample in the study

The study sample was a subset of households from the full SLDHS 2020 dataset that met the inclusion criteria: (1) identified as nomadic or semi-nomadic based on the survey's classification, (2) reported owning camels, and (3) provided complete data on the primary outcome and predictor variables. The SLDHS utilized a multi-stage stratified cluster sampling methodology to ensure regional and national representativeness. After applying the inclusion criteria, the final analytical sample for this study consisted of 6,888 eligible households. The household was the unit of analysis.

## Study variables

The primary outcome variables were the counts of camel mortality reported by each household within the survey's recall period. These are inherently count data. Four specific outcome variables were analyzed: the total number of camels reported dead, the number of camels that died specifically due to drought, the number that died due to floods, and the number that died due to disease. Independent (predictor) variables were selected based on their theoretical relevance and availability in the SLDHS dataset. These were categorized into: 1) Demographic factors: gender of household head, age of household head (categorized), household size (categorized), and education level of the household head; 2) Socioeconomic factors: household wealth quintile; 3) Environmental proxies: primary source of drinking water and Hygiene facility.

## Statistical data analysis

Data analysis was conducted using Stata statistical software (STATA) after data cleaning, preparation, and variable recoding. First, bivariate analysis using the Pearson's Chi-square ($\chi^2$) test for independence was performed to examine the initial, unadjusted associations between each predictor variable and the camel mortality outcomes. To identify the independent determinants of camel mortality while controlling for confounding factors, multivariate count regression models were employed. Given that the outcome variables were non-negative integer counts of events (camel deaths), Negative Binomial Regression was selected as the appropriate analytical approach for modeling count data [12]. Four separate Negative Binomial Regression models were fitted to assess the factors associated with mortality due to drought, flood, disease, and all causes combined. The results of the regression models are presented as Incidence Rate Ratios (IRRs) with their corresponding 95% confidence intervals (CIs) and p-values. An assessment for multicollinearity among predictor variables was conducted using Variance Inflation Factors (VIF), with values well below the common threshold of 10 indicating no significant concern. Statistical significance for all inferential tests was set at a p-value of < 0.05.

## Ethical considerations

This study utilized secondary, anonymized data from the 2020 Somaliland Demographic and Health Survey (SLDHS), which had received prior ethical approval from the relevant ethics committees. In the original survey, informed consent was obtained from all participants before data collection. Consent was written for adult participants, and in the case of minors, consent was obtained from their parents or guardians. The DHS program ensured confidentiality by de-identifying household information and randomly displacing geographic coordinates before releasing the dataset for public use. As this analysis involved no direct contact with participants and used only publicly available, anonymized data, no additional ethical clearance or participant consent was required for this secondary analysis. All procedures adhered to established ethical standards, with strict attention to privacy, confidentiality, and responsible reporting of findings.

## Result

### Summary statistics of key variables in camel mortality models

Table 1 provides foundational descriptive statistics for the key variables analyzed in the camel mortality models. The independent variables, categorized into demographic factors (gender, age, household size, education of head), socioeconomic status (wealth quintile), environmental proxies (water source, toilet facility), and geographic location (region of residence), are primarily categorical. For these, the distribution across their respective categories is presented through percentages, offering an immediate insight into the demographic, economic, infrastructural, and spatial characteristics of the 6,888 nomadic and semi-nomadic households sampled in Somaliland. Together, these statistics lay the groundwork for understanding the baseline characteristics of the study population and the prevalence of factors hypothesized to influence camel mortality, enabling subsequent inferential analyses to explore their associations.

**Table 1. Summary Statistics of Key Variables in Camel Mortality Models (Somaliland, 2020).**

| Variable Category | Variable | Type of Variable | Summary Statistic |
|---|---|---|---|
| **Demographic Factors** | Gender of Household Head | Categorical | Male (78.3%), Female (21.7%) |
| | Age of Household Head | Categorical | Per age group; 25 or less (4.8%), 26–35 (20.3), 36–45 (25.5), 46–55 (21.7), 56 or above (27.9%) |
| | Household Size | Categorical | Per size group; 1–3 members (10.1%), 4–7 members (58.3) 8+ members (31.6%) |
| | Education Level of Household Head | Categorical | Educated (10.1%), Not Educated (89.9%) |
| **Socioeconomic Factors** | Household Wealth Quintile | Categorical | Per quintile; e.g., Poorer (97.2%), Middle (2.7%) |
| **Environmental Proxies** | Primary Source of Drinking Water | Categorical | Per source, e.g., Safely Managed (22.9%), Limited (22.5%), Surface Water (10.1) and Unimproved (44.5%) |
| | Hygiene Facility | Categorical | Per type; No Facility (21.7%), Limited (78.3%) |
| **Geographic Factors** | Region of Residence | Categorical | Per region; Awdal (10.6%), Sanaag (22.1%), Sool (26.8%), Togdheer (18.5%), Maroodijeh (10.9%) and Sahil (10.8%) |

### Bivariate analysis of sociodemographic and environmental determinants of camel mortality

Table 2 presents a strong, statistically significant association that was found between the region of residence and total camel mortality ($x^2$=738.45, p<0.001). The proportion of households experiencing any camel death was highest in Awdal (29.3%) and Sanaag (24.1%), while Togdheer had a remarkably low rate (4.0%). These stark regional disparities were also statistically significant for each specific cause of death (drought, flood, and disease), underscoring the critical need for spatial analysis to target interventions. The age of the household head showed a clear, significant trend with total mortality (($x^2$=281.69, p<0.001). The risk of experiencing camel loss increased steadily with the age of the household head, from just 6.8% for those 25 or less to 19.9% for those 56 or older. The gender of the household head also showed a significant association (($x^2$=66.96, p<0.001), with female-headed households reporting slightly higher overall mortality (16.1% vs 15.5%) and significantly higher mortality from disease.

Notably, the education of the household head did not have a statistically significant relationship with total camel mortality (($x^2$=25.16, p=0.067) at the bivariate level, though it was significantly associated with deaths from drought and disease individually. Household size was significantly associated with total mortality (($x^2$=170.10, p<0.001). Mid-sized households (4–7 members) reported the highest proportion experiencing mortality (17.7%). Surprisingly, the wealth quintile was not significantly associated with total camel mortality (($x^2$=15.24, p=0.507). Although poorer households had a higher rate of mortality (16.2%) compared to middle-income households (5.4%), this difference was not statistically significant for the overall outcome, likely due to the small sample size of the middle-income group. Wealth was, however, significantly associated with drought-related mortality.

Both access to hygiene facilities and water sources were highly significant determinants of total camel mortality. Households with no hygiene facility had a lower proportion of mortality (12.6%) compared to those with limited facilities (16.8%), a counter-intuitive result (($x^2$=150.77, p<0.001) that may be confounded by other factors like region or wealth and requires exploration in the multivariate model. The source of drinking water demonstrated a clear and highly significant gradient of risk (($x^2$=221.57, p<0.001). The proportion of households experiencing camel mortality was lowest for those with safely managed water (12.0%) and highest for those using unimproved sources (19.5%). This reinforces the critical link between water quality, environmental contamination, and livestock health in these nomadic communities.

### Negative binomial regression analysis

**Negative binomial regression of camel die due to drought.** Table 3 presents the results of a negative binomial regression modeling factors influencing the number of camels that died due to drought. The significant coefficients reveal

**Table 2. Bivariate analysis of sociodemographic and environmental determinants of camel mortality, with Chi-square test for association with Total Camel Mortality.**

| Characteristic | Total House-holds n (%) | Camel Mortality due to Drought n (%) | Camel Mortality due to Floods n (%) | Camel Mortality due to Disease n (%) | Total Camel Mortality n (%) | Chi-square ($x^2$) (p-value) |
|---|---|---|---|---|---|---|
| **Region of residence** | | | | | | 738.45 (<0.001)* |
| Awdal | 717 (10.6) | 203 (28.3) | 77 (10.7) | 49 (6.9) | 208 (29.3) | |
| Marodijeh | 732 (10.9) | 160 (21.9) | 9 (1.2) | 43 (5.9) | 74 (10.1) | |
| Sahil | 727 (10.8) | 128 (17.6) | 20 (2.8) | 12 (1.6) | 148 (20.4) | |
| Togdheer | 1,248 (18.5) | 75 (6.0) | 7 (0.6) | 3 (0.2) | 50 (4.0) | |
| Sool | 1,805 (26.8) | 391 (21.7) | 33 (1.8) | 25 (1.4) | 210 (11.6) | |
| Sanaag | 1,486 (22.1) | 474 (31.9) | 48 (3.2) | 49 (3.3) | 358 (24.1) | |
| **Gender of household head** | | | | | | 66.96 (<0.001)* |
| Male | 5,271 (78.3) | 1,184 (22.5) | 144 (2.7) | 100 (1.9) | 818 (15.5) | |
| Female | 1,464 (21.7) | 329 (22.5) | 40 (2.7) | 68 (4.6) | 235 (16.1) | |
| **Education of Household head** | | | | | | 25.16 (0.067) |
| Yes | 681 (10.1) | 177 (26.0) | 22 (3.2) | 25 (3.7) | 114 (16.7) | |
| No | 6,054 (89.9) | 1,328 (21.9) | 163 (2.7) | 154 (2.5) | 957 (15.8) | |
| **Age of household head** | | | | | | 281.69 (<0.001)* |
| 25 or less | 323 (4.8) | 49 (15.2) | 1 (0.3) | 10 (3.1) | 22 (6.8) | |
| 26-35 | 1,357 (20.2) | 206 (15.2) | 15 (1.1) | 6 (0.4) | 122 (9.0) | |
| 36-45 | 1,719 (25.5) | 358 (20.8) | 42 (2.4) | 62 (3.6) | 276 (16.1) | |
| 46-55 | 1,460 (21.7) | 344 (23.6) | 53 (3.6) | 35 (2.4) | 249 (17.1) | |
| 56 or above | 1,876 (27.9) | 511 (27.2) | 73 (3.9) | 52 (2.8) | 374 (19.9) | |
| **Household size** | | | | | | 170.10 (<0.001)* |
| 1-3 | 682 (10.1) | 96 (14.1) | 11 (1.6) | 7 (1.0) | 81 (11.9) | |
| 78.3 | 3,927 (58.3) | 970 (24.7) | 126 (3.2) | 123 (3.1) | 696 (17.7) | |
| 8 or More | 2,126 (31.6) | 438 (20.6) | 47 (2.2) | 34 (1.6) | 286 (13.4) | |
| **Wealth quintile** | | | | | | 15.24 (0.507) |
| Poorer | 6,550 (97.2) | 1,510 (23.1) | 185 (2.8) | 179 (2.7) | 1,061 (16.2) | |
| Middle | 185 (2.7) | 12 (6.5) | 1 (0.5) | 1 (0.5) | 10 (5.4) | |
| **Hygiene facility** | | | | | | 150.77 (<0.001)* |
| Limited | 5,273 (78.3) | 1,228 (23.3) | 94 (1.8) | 118 (2.2) | 887 (16.8) | |
| No Facility | 1,462 (21.7) | 285 (19.5) | 90 (6.2) | 61 (4.2) | 184 (12.6) | |
| **Water source** | | | | | | 221.57 (<0.001)* |
| Safely Managed | 1,540 (22.9) | 223 (14.5) | 24 (1.6) | 42 (2.7) | 185 (12.0) | |
| Limited | 1,516 (22.5) | 312 (20.6) | 21 (1.4) | 20 (1.3) | 216 (14.3) | |
| Unimproved | 2,997 (44.5) | 724 (24.2) | 127 (4.2) | 117 (3.9) | 585 (19.5) | |
| Surface Water | 682 (10.1) | 155 (22.7) | 66 (9.7) | 0 (0.0) | 86 (12.6) | |

several key insights. Older household heads (26−35, 4−7, and 8 or more members, with coefficients of −1.059, −1.084, and −1.287 respectively, all highly significant at p < 0.01) are associated with a significantly lower number of camel deaths. This suggests that households with older heads and larger number of household members might possess more experience in drought management or have better coping strategies. Attending school (coefficient of 0.383, p < 0.05) is associated with an increased number of camel deaths, which is a counter-intuitive finding that might warrant further investigation, potentially indicating that households with schooled members might be less involved in traditional livestock care. Furthermore, wealth quintiles (Middle: −3.089, p < 0.01) show that wealthier households experience significantly

**Table 3. Negative Binomial regression of camel dies due to droughts.**

| Camel Died Due to Drought | Coef. | St.Err. | t-value | p-value | [95% Conf | Interval] | Sig |
|---|---|---|---|---|---|---|---|
| **Age of Household head** | 0 | . | . | . | . | . | |
| 26 - 35 | −1.059 | .263 | −4.03 | 0 | −1.574 | −.543 | *** |
| 36 - 45 | .005 | .266 | 0.02 | .986 | −.517 | .526 | |
| 46 - 55 | −.227 | .268 | −0.85 | .396 | −.752 | .297 | |
| 56 or above | .254 | .263 | 0.96 | .335 | −.262 | .769 | |
| **Sex of household** | 0 | . | . | . | . | . | |
| Female | −.086 | .111 | −0.77 | .439 | −.304 | .132 | |
| **Household Size** | 0 | . | . | . | . | . | |
| 4 - 7 | −1.084 | .169 | −6.40 | 0 | −1.416 | −.752 | *** |
| 8 or More | −1.287 | .188 | −6.85 | 0 | −1.655 | −.919 | *** |
| **Attended School** | 0 | . | . | . | . | . | |
| No | .383 | .152 | 2.52 | .012 | .085 | .682 | ** |
| **Region** | 0 | . | . | . | . | . | |
| Marodijeh | −.156 | .196 | −0.80 | .426 | −.539 | .228 | |
| Sahil | .72 | .208 | 3.46 | .001 | .313 | 1.128 | *** |
| Togdheer | −1.005 | .186 | −5.41 | 0 | −1.368 | −.641 | *** |
| Sool | .297 | .168 | 1.77 | .077 | −.032 | .627 | * |
| Sanaag | 3.23 | .167 | 19.37 | 0 | 2.903 | 3.557 | *** |
| **Wealth quintile** | 0 | . | . | . | . | . | |
| Middle | −3.089 | .385 | −8.03 | 0 | −3.844 | −2.335 | *** |
| **Water Source** | 0 | . | . | . | . | . | |
| Limited | −.185 | .149 | −1.25 | .212 | −.477 | .106 | |
| Unimproved | 1.128 | .134 | 8.41 | 0 | .865 | 1.391 | *** |
| Surface Water | −.236 | .185 | −1.28 | .202 | −.598 | .126 | |
| **Hygiene Facility** | 0 | . | . | . | . | . | |
| No Facility | −.078 | .123 | −0.64 | .524 | −.318 | .162 | |
| Constant | −.119 | .325 | −0.37 | .715 | −.757 | .519 | |
| Inalpha | 2.277 | .031 | .b | .b | 2.216 | 2.338 | |
| Mean dependent var | 7.578 | | SD dependent var | | 82.764 | | |
| Pseudo r-squared | 0.106 | | Number of obs | | 6888 | | |
| Chi-square | 1764.169 | | Prob > chi2 | | 0.000 | | |
| Akaike crit. (AIC) | 14955.659 | | Bayesian crit. (BIC) | | 15092.409 | | |

*** p < .01, ** p < .05, * p < .1.

fewer camel deaths. Regional variations are also pronounced, with Sahil (0.72, p < 0.01) and Sanaag (3.23, p < 0.01) showing significantly more camel deaths, while Togdheer (−1.005, p < 0.01) indicates fewer. Access to unimproved water sources (1.128, p < 0.01) is strongly linked to more camel deaths, highlighting the critical role of water infrastructure.

**Negative binomial regression of camel die due to flood.** Table 4 examines the factors influencing camel deaths due to floods. Contrary to drought, older household heads (4−7: −2.779, p < 0.01; 8 or more: −3.582, p < 0.01) are again associated with significantly fewer camel deaths, implying their experience also aids in flood mitigation or preparedness. Interestingly, being female (coefficient of 0.08, p < 0.01) is associated with a significantly lower number of camel death, which might reflect gendered roles in flood response or access to resources. Regionally, Marodijeh (−5.505, p < 0.01), Togdheer (−2.341, p < 0.01), and Sool (−2.525, p < 0.01) show significantly fewer camel deaths, while Sanaag (1.921,

**Table 4. Negative Binomial Regression of Camel Die due to Flood.**

| Camel Died Due to Floods | Coef. | St.Err. | t-value | p-value | [95% Conf | Interval] | Sig |
|---|---|---|---|---|---|---|---|
| **Age of Household head** | 0 | . | . | . | . | . | |
| 26 - 35 | −.698 | 1.021 | −0.68 | .494 | −2.699 | 1.302 | |
| 36 - 45 | 1.252 | 1.138 | 1.10 | .271 | −.979 | 3.483 | |
| 46 - 55 | 2.774 | 1.11 | 2.50 | .012 | .599 | 4.949 | ** |
| 56 or above | 1.535 | 1.155 | 1.33 | .184 | −.729 | 3.799 | |
| **Sex of household** | 0 | . | . | . | . | . | |
| Female | −.08 | .399 | −0.20 | .841 | −.862 | .702 | |
| **Household Size** | 0 | . | . | . | . | . | |
| 4 - 7 | −2.779 | .753 | −3.69 | 0 | −4.255 | −1.302 | *** |
| 8 or More | −3.582 | .844 | −4.25 | 0 | −5.236 | −1.929 | *** |
| **Attended School** | 0 | . | . | . | . | . | |
| No | .387 | .549 | 0.70 | .481 | −.689 | 1.463 | |
| **Region** | 0 | . | . | . | . | . | |
| Marodijeh | −5.505 | .983 | −5.60 | 0 | −7.431 | −3.579 | *** |
| Sahil | −.795 | .736 | −1.08 | .28 | −2.238 | .648 | |
| Togdheer | −2.341 | .861 | −2.72 | .007 | −4.029 | −.654 | *** |
| Sool | −2.525 | .733 | −3.44 | .001 | −3.963 | −1.088 | *** |
| Sanaag | 1.921 | .829 | 2.32 | .021 | .296 | 3.547 | ** |
| | 0 | . | . | . | . | . | |
| Middle | .581 | 1.475 | 0.39 | .694 | −2.311 | 3.473 | |
| **Water Source** | 0 | . | . | . | . | . | |
| Limited | −1.265 | .563 | −2.25 | .025 | −2.369 | −.161 | ** |
| Unimproved | 3.235 | .636 | 5.09 | 0 | 1.989 | 4.481 | *** |
| Surface Water | .376 | .668 | 0.56 | .574 | −.934 | 1.685 | |
| **Hygiene Facility** | 0 | . | . | . | . | . | |
| No Facility | 2.136 | .599 | 3.56 | 0 | .961 | 3.31 | *** |
| Constant | −.837 | 1.169 | −0.72 | .474 | −3.128 | 1.453 | |
| Inalpha | 4.508 | .073 | .b | .b | 4.365 | 4.65 | |
| Mean dependent var | 7.995 | | SD dependent var | | 87.904 | | |
| Pseudo r-squared | 0.080 | | Number of obs | | 6888 | | |
| Chi-square | 341.609 | | Prob > chi2 | | 0.000 | | |
| Akaike crit. (AIC) | 3965.870 | | Bayesian crit. (BIC) | | 4102.621 | | |

*** p<.01, ** p<.05, * p<.1.

p<0.05) is associated with more, indicating differential vulnerability across regions. Similar to drought, access to unimproved water sources (3.235, p<0.01) is significantly associated with an increased number of camel deaths, underscoring the vulnerability associated with such sources during floods. Households with no hygiene facility (2.136, p<0.01) also experience significantly more camel deaths, suggesting a link between overall household infrastructure and resilience to flood impacts.

**Negative binomial regression of camel die due to disease.** Table 5 details the factors affecting camel deaths due to disease. Similar to the previous tables, household heads in the younger age groups (26−35: −3.861, p<0.01; 36−45: −2.958, p<0.01; 46−55: −3.025, p<0.01; 56 or above: −2.022, p<0.1) are associated with significantly fewer camel deaths, with the exception of the very youngest group. Being female (1.576, p<0.01) is associated with a significantly

**Table 5. Negative binomial Regression of Camel Die due to disease.**

| Camel Died Due to Disease | Coef. | St.Err. | t-value | p-value | [95% Conf | Interval] | Sig |
|---|---|---|---|---|---|---|---|
| **Age of Household head** | 0 | . | . | . | . | . | |
| 26 - 35 | −3.861 | 1.033 | −3.74 | 0 | −5.885 | −1.836 | *** |
| 36 - 45 | −2.958 | 1.073 | −2.76 | .006 | −5.06 | −.855 | *** |
| 46 - 55 | −3.025 | 1.095 | −2.76 | .006 | −5.171 | −.879 | *** |
| 56 or above | −2.022 | 1.131 | −1.79 | .074 | −4.24 | .195 | * |
| **Sex of household** | 0 | . | . | . | . | . | |
| Female | 1.576 | .415 | 3.80 | 0 | .764 | 2.389 | *** |
| **Household Size** | 0 | . | . | . | . | . | |
| 4 - 7 | −1.172 | .787 | −1.49 | .136 | −2.715 | .37 | |
| 8 or More | −1.878 | .834 | −2.25 | .024 | −3.512 | −.244 | ** |
| **Attended School** | 0 | . | . | . | . | . | |
| No | 1.914 | .577 | 3.32 | .001 | .782 | 3.045 | *** |
| **Region** | 0 | . | . | . | . | . | |
| Marodijeh | −.209 | .707 | −0.30 | .768 | −1.595 | 1.177 | |
| Sahil | −.454 | .758 | −0.60 | .549 | −1.939 | 1.031 | |
| Togdheer | −2.142 | .828 | −2.59 | .01 | −3.765 | −.518 | *** |
| Sool | −1.559 | .656 | −2.38 | .017 | −2.843 | −.274 | ** |
| Sanaag | 5.215 | .523 | 9.97 | 0 | 4.19 | 6.24 | *** |
| | 0 | . | . | . | . | . | |
| Middle | −4.532 | 1.896 | −2.39 | .017 | −8.248 | −.816 | ** |
| **Water Source** | 0 | . | . | . | . | . | |
| Limited | −2.279 | .535 | −4.26 | 0 | −3.326 | −1.231 | *** |
| Unimproved | .241 | .582 | 0.41 | .679 | −.9 | 1.382 | |
| Surface Water | −29.283 | 14313.596 | −0.00 | .998 | −28083.415 | 28024.849 | |
| **Hygiene Facility** | 0 | . | . | . | . | . | |
| No Facility | .391 | .465 | 0.84 | .401 | −.521 | 1.302 | |
| Constant | .256 | 1.266 | 0.20 | .84 | −2.226 | 2.737 | |
| lnalpha | 4.368 | .075 | .b | .b | 4.221 | 4.514 | |
| Mean dependent var | 7.878 | | SD dependent var | | | 87.756 | |
| Pseudo r-squared | 0.099 | | Number of obs | | | 6888 | |
| Chi-square | 409.132 | | Prob > chi2 | | | 0.000 | |
| Akaike crit. (AIC) | 3749.648 | | Bayesian crit. (BIC) | | | 3886.399 | |

*** p<.01, ** p<.05, * p<.1.

higher number of camel deaths, a consistent finding across flood and disease related deaths, suggesting a systemic vulnerability. The middle wealth quintile (−4.532, p<0.05) indicates fewer camel deaths, reinforcing the protective effect of wealth. Regionally, Togdheer (−2.142, p<0.01), Sool (−1.559, p<0.05), and Marodijeh (−0.209, not significant) show fewer camel deaths, while Sanaag (5.215, p<0.01) is associated with significantly more disease-related deaths, again highlighting regional disparities in disease prevalence or control. Limited water sources (−2.279, p<0.01) are linked to significantly fewer disease-related deaths, a potentially complex finding that might relate to reduced exposure to pathogens in certain limited sources or different watering practices.

**Negative binomial regression for total camel deaths.** Table 6 aggregates the factors influencing the total number of camel deaths from all causes. Consistent with previous findings, younger age groups for household heads (26−35: −1.723,

**Table 6. Negative binomial Regression for Total Camel Deaths.**

| Total Camel Died | Coef. | St.Err. | t-value | p-value | [95% Conf | Interval] | Sig |
|---|---|---|---|---|---|---|---|
| **Age of Household head** | 0 | . | . | . | . | . | |
| 26 - 35 | −1.723 | .29 | −5.94 | 0 | −2.291 | −1.154 | *** |
| 36 - 45 | −.43 | .301 | −1.43 | .153 | −1.02 | .16 | |
| 46 - 55 | −1.615 | .293 | −5.51 | 0 | −2.189 | −1.041 | *** |
| 56 or above | −.83 | .288 | −2.88 | .004 | −1.395 | −.265 | *** |
| **Sex of household** | 0 | . | . | . | . | . | |
| Female | −.385 | .121 | −3.18 | .001 | −.621 | −.148 | *** |
| **Household Size** | 0 | . | . | . | . | . | |
| 4 - 7 | −.051 | .199 | −0.25 | .799 | −.44 | .339 | |
| 8 or More | −.653 | .216 | −3.03 | .002 | −1.076 | −.23 | *** |
| | 0 | . | . | . | . | . | |
| No | .036 | .155 | 0.24 | .814 | −.267 | .34 | |
| **Region** | 0 | . | . | . | . | . | |
| Marodijeh | −1.271 | .207 | −6.15 | 0 | −1.676 | −.866 | *** |
| Sahil | .496 | .222 | 2.23 | .026 | .061 | .932 | ** |
| Togdheer | −.811 | .195 | −4.16 | 0 | −1.194 | −.429 | *** |
| Sool | −.522 | .175 | −2.99 | .003 | −.864 | −.18 | *** |
| Sanaag | .22 | .181 | 1.22 | .223 | −.134 | .575 | |
| | 0 | . | . | . | . | . | |
| Middle | −2.887 | .423 | −6.82 | 0 | −3.717 | −2.058 | *** |
| **Water Source** | 0 | . | . | . | . | . | |
| Limited | −.999 | .148 | −6.76 | 0 | −1.288 | −.709 | *** |
| Unimproved | .125 | .127 | 0.99 | .324 | −.124 | .374 | |
| Surface Water | −1.215 | .189 | −6.43 | 0 | −1.585 | −.845 | *** |
| **Hygiene Facility** | 0 | . | . | . | . | . | |
| No Facility | .282 | .126 | 2.23 | .026 | .034 | .529 | ** |
| Constant | 1.338 | .346 | 3.87 | 0 | .661 | 2.016 | *** |
| Inalpha | 2.393 | .038 | .b | .b | 2.317 | 2.468 | |
| Mean dependent var | 1.017 | | SD dependent var | | | 19.114 | |
| Pseudo r-squared | 0.049 | | Number of obs | | | 6888 | |
| Chi-square | 554.694 | | Prob > chi2 | | | 0.000 | |
| Akaike crit. (AIC) | 10880.305 | | Bayesian crit. (BIC) | | | 11017.056 | |

*** p<.01, ** p<.05, * p<.1.

p<0.01; 46−55: −1.615, p<0.01; 56 or above: −0.83, p<0.01) are associated with significantly fewer camel deaths, while the 36−45 group is not significant in this combined analysis. Being female (−0.385, p<0.01) is associated with fewer total camel deaths, which contradicts the individual flood and disease tables and may point to a different overarching dynamic. This suggests that while females might be more vulnerable to specific events like floods and disease, their overall herd management practices might be more protective against total mortality. The middle wealth quintile (−2.887, p<0.01) is again linked to significantly fewer camel deaths, confirming the protective effect of higher wealth. Regionally, Marodijeh (−1.271, p<0.01), Togdheer (−0.811, p<0.01), and Sool (−0.522, p<0.01) are associated with fewer total camel deaths, while Sahil (0.496, p<0.05) shows more, further underscoring the importance of geographic context. Limited water sources (−0.999, p<0.01) are associated with significantly fewer total camel deaths, echoing the disease-specific finding.

Finally, having no hygiene facility (0.282, p<0.05) is associated with significantly more total camel deaths, highlighting the importance of basic sanitation infrastructure in overall herd health and resilience.

**Assessment of multicollinearity using variance inflation factors (VIF).** Table 7 presents an assessment of Multicollinearity among the predictor variables was conducted using Variance Inflation Factors (VIF). All VIF scores were substantially low, ranging from 1.012 for school attended to 1.094 for Total number of household members. The mean VIF was 1.051. As all individual VIF values are well below the common threshold of 5 (or 10), and the mean VIF is very close to 1, Multicollinearity is not considered a concern for this set of predictors in the model.

**Summary of major findings.** Fig 1 illustrates the regional heterogeneity and etiological stratification of camel mortality in Somaliland, classifying losses due to drought, floods, and disease, along with overall death rate. The Geographical study reveals that drought is the primary cause of camel loss, demonstrating the most significant impact, especially in Sanaag and Awdal regions, which reported mortality rates of 31% and 28.3%, respectively. Although death from floods and diseases adds less to overall statistics, certain localized vulnerabilities are apparent; particularly, Awdal stands out as a critical area for flood-related mortality (10.7%) and disease-associated losses (6.9%). The distribution of total camel mortality reveals a significant regional disparity, with the peripheral regions of Awdal and Sanaag experiencing the hisghest cumulative attrition, in stark contrast to the central region Togdheer, which consistently has the lowest mortality rates across all variables categories,

## Discussion

This study provides a comprehensive analysis of the socio-economic and environmental determinants of camel mortality in Somaliland's nomadic communities, utilizing household level data from the 2020 Demographic and Health Survey. Our findings indicate that camel populations, despite their inherent resilience, are significantly impacted by a complex interplay of factors, consistent with broader trends observed in arid and semi-arid regions where livestock face considerable environmental challenges [1,2]. The observed high calf mortality rates, a major constraint highlighted in this research, align with previous reports from other East African settings [2,4,13]. This particular vulnerability in young stock underscores the urgent need for targeted interventions focusing on early-life management and disease prevention. Such efforts should build upon existing ethno-veterinary practices while integrating modern approaches to enhance overall herd sustainability [14].

A key finding of this research is the pronounced regional disparity in camel mortality. Our bivariate analysis revealed a strong and statistically significant association between the region of residence and total camel mortality. Areas such as Awdal and Sanaag consistently reported higher rates of camel deaths across all categories (drought, flood, and disease), in stark contrast to Togdheer, which demonstrated remarkably lower mortality. This geographic variation underscores the critical importance of spatial analysis in designing effective interventions [10,15] . These regional differences likely reflect variations in environmental susceptibility, resource availability, access to veterinary services, and prevailing socio-economic conditions, demanding region-specific strategies rather than a uniform approach. Studies on camel production

**Table 7. Variance inflation factor.**

| Variable | VIF | 1/VIF |
|---|---|---|
| household members | 1.094 | 0.914 |
| Sex of Household head | 1.061 | 0.942 |
| Hygiene Facility | 1.059 | 0.944 |
| Region | 1.057 | 0.946 |
| Water Facility | 1.046 | 0.956 |
| Wealth | 1.030 | 0.971 |
| School Attendance | 1.012 | 0.988 |
| Mean VIF | 1.051 | |

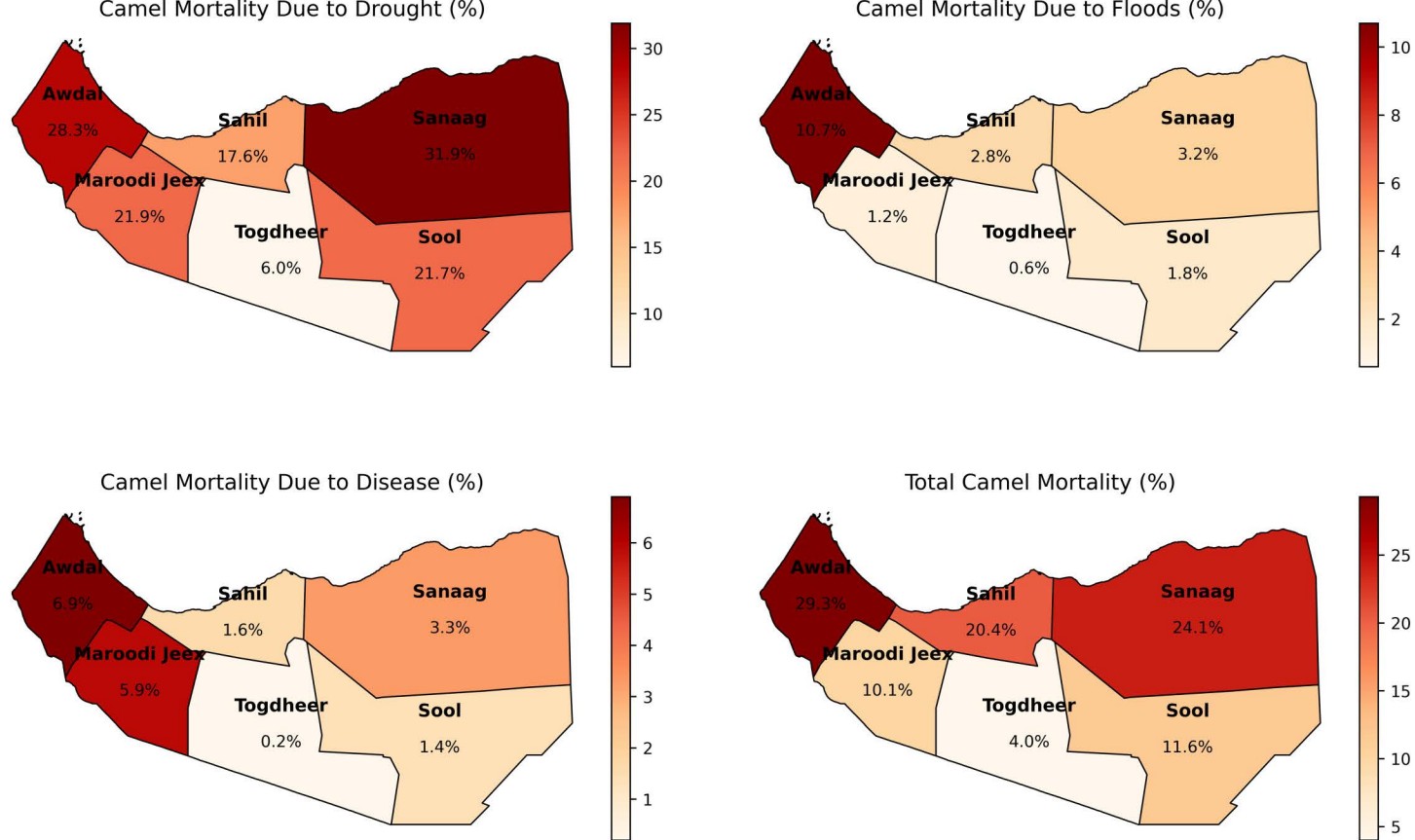

**Fig 1. Regional Distribution of Camel Mortality Rates in Somaliland.** Administrative boundary data used to generate this map were obtained from the Global Administrative Areas database (GADM; https://gadm.org/) and are used under the Creative Commons Attribution 4.0 International License (CC BY 4.0).

in Pakistan have similarly noted significant regional differences in management practices and challenges, indicating a shared need for geographically tailored solutions in pastoral communities (Faraz et al., 2019; Faraz et al., 2020). The Omani study also highlighted that camel farming is a male-dominated activity, and the primary source of revenue from camel racing could vary by region, influencing investment in camel health and management [16].

Socio-demographic factors of household heads, such as age and gender, exhibited significant associations with camel mortality, though these impacts varied across different causes of death. Older household heads and those with larger household sizes were linked to significantly fewer camel deaths, particularly due to drought and floods. This suggests that accumulated experience in traditional drought management and coping strategies within these demographics, combined with greater labor resources, may confer greater resilience, a point consistent with the importance of indigenous knowledge in pastoral communities [17,18]. Conversely, female-headed households consistently experienced higher camel mortality due to disease and floods, hinting at potential gendered vulnerabilities in resource access or flood response mechanisms. This finding aligns with the call for gender training to strengthen camel husbandry practices, as suggested by [14] in the Pakistani context, and highlights a complex dynamic where specific vulnerabilities might exist despite overall protective management practices. Omani study, in noting camel farming as male-dominated, further contextualizes potential differences in management and resource allocation [16].

Access to unimproved water sources was identified as a strong determinant of increased camel mortality, particularly during droughts and floods, underscoring the critical link between water infrastructure, environmental contamination, and livestock health. This finding echoes concerns about water supply as a major constraint on camel production [19,20]. Furthermore, the absence of a hygiene facility was associated with significantly more total camel deaths, suggesting a direct link between basic sanitation infrastructure and overall herd health and resilience. This reinforces the need for proper hygiene and disease prevention, a point also made by the Omani study which suggested better medical support and periodic inspections to improve camel health [16]. The counter-intuitive finding that households with no hygiene facilities had lower overall mortality in one instance warrants further investigation, as it may be confounded by other factors such as wealth or regional location, necessitating a deeper multivariate analysis. Moreover, Somaliland's unique climate further shapes the underlying pattern of risk, prolonged dry season impose severe nutritional constraints, whereas rainy periods facilitate disease spread through herd aggregation and contamination of water source.

This study reinforces the understanding that camel mortality in nomadic communities is a complex issue driven by the interplay of environmental stressors, disease prevalence, and socio-economic vulnerabilities. While traditional ethno-veterinary practices play a crucial role in primary healthcare [14,18], the lack of market investments, limited access to formal veterinary services, and poor infrastructure continue to be major constraints, as similarly observed in Pakistani pastoral farming [14]. The findings advocate for integrated interventions that combine traditional knowledge with modern veterinary support, alongside improved water and hygiene infrastructure, and targeted educational and economic empowerment programs.

Additionally, Somali pastoralists traditionally practice cauterization to treat lameness and chronic disease. In many cases they also employ sterile techniques and post-procedural pain management to reduce the risk of secondary infection, practices that could be further aligned with the modern veterinary standards. Moreover, the use of phytotherapeutic remedies, which are low cost, offers additional potential for controlling internal parasites and wound healing. Such a multi-faceted approach, including interest-free small loans and community education, is essential to strengthen the camel value chain, improve breeders' quality of life, and enhance food security in these climate-vulnerable regions [14]. The Omani research's suggestions for strong fencing, private veterinary hospitals, and government controls on clinic prices also align with the need for enhanced infrastructure and support to mitigate camel fatalities [16].

In the Somaliland context, future research should prioritize longitudinal panel studies and meteorological and geospatial data, while future data collection would capture seasonal granularity wet and dry seasons. This would help to track nomadic households and their herds over multiple times and provide the region's actual climate calendar. In addition to improving temporal data, future analytical frameworks must also incorporate the multidimensional aspects of risk. In the current analysis, Environmental and demographic factors are treated as separate, additive sources of risks. However, this approach obscures potential interaction effects that may generate exacerbated vulnerabilities with significant implications for targeting interventions. In particular, investigating the relationship between region and water source may reveal that mortality rates associated with unimproved water are disproportionately high in eastern regions such as Sanaag and Sool compared with the west. This would parallel findings for Ethiopian pastoral systems, where drought-related livestock losses tend to cluster around limited water sources [21].

## Conclusion

This study successfully explored the magnitude and determinants of camel mortality among nomadic communities in Somaliland, revealing that camel losses are systematically shaped by a confluence of household demographics, and environmental conditions. The primary determinants of increased mortality risk were residence in the eastern regions of Somaliland, older age of the household head, and reliance on unimproved water sources. The findings unequivocally show that environmental shocks, particularly drought, are the most significant drivers of camel mortality, underscoring the profound vulnerability of pastoral livelihoods to climate variability. The most powerful conclusion drawn from this research

is the existence of a clear and statistically significant east-west in camel survival. The eastern regions of Togdheer, Sool, and Sanaag constitute a major mortality, representing the epicenter of camel loss from combined causes. Conversely, the western region of Awdal is a resilient of a low mortality. By leveraging a large-scale demographic survey for veterinary epidemiological analysis, this study provides an evidence-based foundation for policy and practice. The findings suggest that infrastructure investments, particularly to support communities transitioning away from unimproved water sources, should be prioritized by the Ministry of Livestock and Rural Development and its partners. In particular, efforts should focus on protecting boreholes and sanitation facilities and mitigating disease vectors in the eastern parts of Sanaag, Sool, and Togdheer. In addition, mobile veterinary clinics and subsidized drug distribution networks should be established with the support of international NGOs and local pastoral associations. Together, these measures clearly indicate where and for whom interventions are most urgently needed to safeguard this vital livelihood asset and strengthen pastoralist resilience in the face of mounting environmental pressure.

## Author contributions

**Conceptualization:** Mohamed Ahmed Hassan.

**Formal analysis:** Omran Salih, Mohamed Ahmed Hassan.

**Investigation:** Omran Salih, Mohamed Ahmed Hassan.

**Software:** Mohamed Ahmed Hassan.

**Supervision:** Omran Salih.

**Writing – original draft:** Mohamed Ahmed Hassan.

**Writing – review & editing:** Omran Salih.

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
