## [Decision Letter · Decision Letter 0]

16 Sep 2025

PONE-D-25-37649Socio-demographic and Environmental Determinants of Camel Mortality in Somaliland's Nomadic Communities: A Count Regression Analysis of the 2020 Demographic and Health SurveyPLOS ONE

Dear Dr. Salih,

Thank you for submitting your manuscript to PLOS ONE. After careful consideration, we feel that it has merit but does not fully meet PLOS ONE’s publication criteria as it currently stands. Therefore, we invite you to submit a revised version of the manuscript that addresses the points raised during the review process.

**ACADEMIC EDITOR:** 

Dear authors,

The reviewers' advise is to undertake extensive revisions to improve the content of your manuscript. Please revise your manuscript to address each point raised by them and answer their questions. Please submit the manuscript together with a file containing responses to reviewer comments.

Note from Editorial Staff: Please note that Reviewer 3 has provided their comments as an attached Word document only, due to to technical issues. This file is titled "Socio-demographic and Environmental Determinants_Reviewer 3.docx". Please address the comments from all three reviewers in your revision.

We look forward to receiving your revised manuscript.

Kind regards,

Nussieba A. Osman, Dr. Med. Vet.

Academic Editor

PLOS ONE

Journal Requirements:

Additional Editor Comments:

Reviewer #3: the file is attached.

Reviewers' comments:

Reviewer's Responses to Questions

**Comments to the Author**

1. Is the manuscript technically sound, and do the data support the conclusions?

Reviewer #1: Yes

Reviewer #2: Partly

2. Has the statistical analysis been performed appropriately and rigorously? 

Reviewer #1: Yes

Reviewer #2: Yes

3. Have the authors made all data underlying the findings in their manuscript fully available?

Reviewer #1: Yes

Reviewer #2: Yes

4. Is the manuscript presented in an intelligible fashion and written in standard English?

Reviewer #1: Yes

Reviewer #2: Yes

5. Review Comments to the Author

Reviewer #1: Congratulations for such a nice work. Kindly add the suggested changes in the text. The results should be compared with the latest paper mentioned in the text. That would make the paper more presentable and comparable.

Reviewer #2: Dear Authors,

Thank you for the opportunity to review your manuscript titled "Socio-demographic and Environmental Determinants of Camel Mortality in Somaliland's Nomadic Communities: A Count Regression Analysis of the 2020 Demographic and Health Survey." Your work provides a valuable contribution to understanding how social, demographic, and environmental factors influence camel mortality in pastoral communities. The use of nationally representative data and the application of multivariate count regression models are commendable and well-suited to the study's objectives. As a reviewer, I appreciate your efforts in addressing this important topic. I also have some comments and questions, which I hope will help improve the clarity, transparency of methods, and practical relevance of your manuscript.

Major comments

1- Variable definitions and interpretations:

The term “disease” is used broadly throughout the document. Could you clarify which specific conditions or syndromes this includes in your context? If disease is considered an outcome variable, I suggest adding a section that reviews the major diseases threatening camel productivity in the region to provide perspective for your readers and to address whether these diseases can occur concurrently alongside drought or flood. This will be relevant for the econometrics and validity of your model.

The result indicating lower incidence rates in female-headed households (IRR = 0.97) seems inconsistent with earlier findings of higher mortality in such households. Please clarify this apparent contradiction and consider the conditions under which the effect direction might change.

2- Methodological issues:

Poisson regression assumes equi-dispersion. Please provide evidence that the mean and variance of your outcome variables are equal; otherwise, I suggest considering a Negative Binomial model if overdispersion does exists.

Given the interconnectedness of drought, water scarcity, and disease transmission, how did you address the independence assumption of the count models?

Can you provide a table with summary statistics of all relevant variables included in the models before the bivariate analysis in Table 1? Also, clarify what the values in brackets in Table 1 represent.

3- Operationalization of variables:

Why were the age of the household head and household size both categorized rather than treated as continuous variables? Similarly, why is education dichotomized rather than represented by levels of completion as commonly framed in similar studies?

Please explain why household water sources and toilet facilities are used as proxies for environmental variables in a camel mortality study. Wouldn't environmental and biophysical indicators such as temperature or heat, rainfall variability, green pasture condition, or access to water points be better environmental proxies?

4- Interpretation of results:

The finding that mid-sized households (up to 7 members) reported the highest camel mortality is intriguing but not discussed in sufficient detail. Could you elaborate on potential socioeconomic or logistical reasons?

Similarly, you found that households without hygiene facilities had lower camel mortality rates than those with some hygiene facilities. Could you critically examine this counterintuitive finding and discuss its implications for environmental health metrics?

Regarding herd size, it was assumed that larger herds imply more wealth and management resources. Why, then, would bigger herds be associated with higher mortality? Isn’t this related to increased disease susceptibility or still drought-related risks in higher-density herds?

5- Contextualization and recommendations:

Consider comparing your findings with those of similar studies from other countries or regions with comparable pastoral systems. It would be valuable to compare and contrast policy interventions that solely address climatic shocks, as you recommended, with those that combine or bundle climatic and health risk management and report on their various outcomes.

Also, outline a more specific plan for implementing your recommendations, including expected results and potential barriers to implementation.

Again, thank you for your important work. I hope these comments and questions help improve your manuscript and enhance its contribution to the field of livestock management in pastoral settings.

Sincerely,

Reviewer

6. PLOS authors have the option to publish the peer review history of their article (what does this mean? ). If published, this will include your full peer review and any attached files.

**Do you want your identity to be public for this peer review?** For information about this choice, including consent withdrawal, please see our Privacy Policy .

Reviewer #1: **Yes:** Asim Faraz

Reviewer #2: No

---

## [Author Response · Author response to Decision Letter 1]

9 Oct 2025

Manuscript ID: SO-25-2570

Manuscript Title: Socio-demographic and Environmental Determinants of Camel Mortality in Somaliland's Nomadic Communities: A Count Regression Analysis of the 2020 Demographic and Health Survey

To: PLoS ONE

Re: Response to reviewers

Dear Respected Editor,

Thank you for allowing the resubmission of our manuscript, with an opportunity to address the reviewers’ comments.

We are uploading (a) our point-by-point response to the comments (below) (response to reviewers), (b) an updated manuscript with green highlighting indicating changes, and (c) a clean updated manuscript without highlights.

Finally, we would like to thank the reviewers for reviewing the paper and providing suggestions to improve the paper’s quality. The authors have addressed the comments and suggestions, as described in the response below.

We look forward to your decision.

Best regards,

The paper’s authors.

Reviewer #1

Reviewer #1, Concern #1: Congratulations for such nice work. Kindly add the suggested changes in the text. The results should be compared with the latest paper mentioned in the text. That would make the paper more presentable and comparable.

Authors response to Reviewer #1, Concern #1: We thank the reviewer for this valuable comment. We have compared the results in latest papers mentioned in the text and kindly see the discussion section (See Page 18).

Reviewer #2

Reviewer #2, Concern #1: The term “disease” is used broadly throughout the document. Could you clarify which specific conditions or syndromes this includes in your context? If disease is considered an outcome variable, I suggest adding a section that reviews the major diseases threatening camel productivity in the region to provide perspective for your readers and to address whether these diseases can occur concurrently alongside drought or flood. This will be relevant for the econometrics and validity of your model.

Authors response to Reviewer #2, Concern #1: We thank the reviewer for this valuable comment. In the revised manuscript, we have substantially mentioned the major diseases threatening to camel productively.

The word “disease” is considered an outcome variable, but we have added the major diseases threatening camel productivity in the region. In the Somali region of Ethiopia, consistent with Somaliland, livestock keepers identified Camelpox as the top disease cause of camel calf mortality with a mean score of 4.75, followed by Calf Scour at 4.08. Contagious ecthyma, Tick, and Pneumonic Diseases ranked as the third, fourth, and fifth leading causes of mortality, respectively. (Hussein et al., 2024). (See Page 2)

Reviewer #2, Concern #2: The result indicating lower incidence rates in female-headed households (IRR = 0.97) seems inconsistent with earlier findings of higher mortality in such households. Please clarify this apparent contradiction and consider the conditions under which the direction of effect might change.

Authors response to Reviewer #2, Concern #2: After we applied the Negative Binomial regression analysis it shows that female-headed households had a slightly lower expected number of camel deaths due to drought compared to male-headed households (Coef. = -0.086). However, this effect was not statistically significant (p = 0.439), and the 95% confidence interval (-0.304 to 0.132) includes zero, indicating substantial uncertainty around the estimate. Therefore, household head gender does not appear to significantly influence camel mortality during drought events in this study population (Page 10).

Reviewer #2, Concern #2: Poisson regression assumes equi-dispersion. Please provide evidence that the mean and variance of your outcome variables are equal; otherwise, I suggest considering a Negative Binomial model if over-dispersion does exists.

Authors response to Reviewer #2, Concern #2: We sincerely thank the reviewer for this important observation. In response, We carefully checked the assumptions of equi-distribution, over-dispersion, and under-dispersion, found significant over-dispersion, and subsequently applied negative binomial regression (Page 10-17).

Reviewer #2, Concern #3: Given the interconnectedness of drought, water scarcity, and disease transmission, how did you address the independence assumption of the count models?

Authors response to Reviewer #2, Concern #3: We primarily addressed the independence assumption in our count models by employing Negative Binomial regression, which robustly handles over-dispersion. Furthermore, we mitigated potential dependencies by incorporating regional categorical variables into our multivariate analyses.

Reviewer #2, Concern #4: Can you provide a table with summary statistics of all relevant variables included in the models before the bivariate analysis in Table 1? Also, clarify what the values in brackets in Table 1 represent.

Authors response to Reviewer #2, Concern #4: We have provided a table with summary statistics of all relevant variables included in the models before the bivariate analysis in Table 1 (Page 6). Regarding the previous Table 1 or current Table 2, the values in brackets represent the percentage (% within each category. For example, under "Region," "Awdal 717 (10.6)" means that 717 households reside in Awdal, which constitutes 10.6% of the total households in the sample. Similarly, "203 (28.3)" under "Camel Mortality due to Drought" for Awdal means that 203 camel deaths due to drought were reported in Awdal, representing 28.3% of the total camel deaths in Awdal due to drought (Page 7-8).

Reviewer #2, Concern #5: Why were the age of the household head and household size both categorized rather than treated as continuous variables? Similarly, why is education dichotomized rather than represented by levels of completion as commonly framed in similar studies?

Please explain why household water sources and toilet facilities are used as proxies for environmental variables in a camel mortality study. Wouldn't environmental and biophysical indicators such as temperature or heat, rainfall variability, green pasture condition, or access to water points be better environmental proxies?

Authors response to Reviewer #1, Concern #5: We appreciate the reviewer’s thoughtful observation. In the variables categorization:

The "age of household head" and "household size" likely stem from the desire to capture non-linear relationships with camel mortality, as the impact may not be uniform across all ages or sizes. For instance, very young or old household heads, or specific household sizes, might face distinct risks. This approach also enhances interpretability and policy relevance, making it easier for stakeholders to understand and act on findings (e.g., "households with heads aged 56+ have a different risk"). Furthermore, it can help manage data distribution issues and outliers, creating more robust analytical groups and simplifying model building. Also, for "education level" dichotomization into "Yes/No" (educated/not educated) was probably driven by data availability and quality from the Somaliland Demographic and Health Survey (SLDHS 2020), which might have collected education in a binary format.

While direct environmental indicators like temperature, rainfall, and pasture conditions would indeed be ideal, the study's reliance on the human-centric Somaliland Demographic and Health Survey (SLDHS 2020) necessitated the use of household water sources and toilet facilities as proxies. The SLDHS primarily collects household-level human health and socioeconomic data, lacking the granular, environmental and veterinary information needed for direct biophysical analysis of livestock. We explicitly acknowledged this data limitation. In nomadic pastoralist settings, household water sources directly reflect livestock's access to water and vulnerability to contamination and drought; unimproved human water sources often imply similar conditions for camels, increasing disease risk. Similarly, household toilet facilities serve as proxies for overall environmental sanitation and pathogen load, indirectly impacting livestock health in shared environments. These choices, though indirect, were pragmatic, allowing the us to leverage available data to infer localized environmental quality, disease exposure, and water stress, bridging the gap between human household characteristics and camel mortality in a context where human and animal well-being are closely intertwined.

Reviewer #2, Concern #6: The finding that mid-sized households (up to 7 members) reported the highest camel mortality is intriguing but not discussed in sufficient detail. Could you elaborate on potential socioeconomic or logistical reasons?

Similarly, you found that households without hygiene facilities had lower camel mortality rates than those with some hygiene facilities. Could you critically examine this counterintuitive finding and discuss its implications for environmental health metrics?

Regarding herd size, it was assumed that larger herds imply more wealth and management resources. Why, then, would bigger herds be associated with higher mortality? Isn’t this related to increased disease susceptibility or still drought-related risks in higher-density herds?

Authors response to Reviewer #2, Concern #6: We thank the reviewer for this suggestion. When we applied the negative binomial regression Mid and large household size associated with a significantly lower number of camel deaths. This suggests that households with larger number of household members might possess more experience in drought management or have better coping strategies (Page 10).

After we have applied the negative binomial regression we have found that households with no hygiene facility (2.136, p<0.01) also experience significantly more camel deaths, suggesting a link between overall household infrastructure and resilience to flood impacts (Page 12).

You're right to question the counter-intuitive finding that larger herds, despite implying more wealth and resources, are associated with higher camel mortality. This likely reflects complex dynamics: larger, denser herds create an ideal environment for rapid disease transmission due to closer contact and increased stress, as well as exacerbating drought-related risks by placing greater demand on finite pasture and water resources, leading to quicker depletion and pushing herders into more marginal areas. Crucially, as the paper suggests, wealthier households often own significantly more camels; therefore, even if their proportional loss during a shock might be lower, the absolute number of deaths in a larger herd can still be higher, driving the mortality count. Furthermore, the sheer scale of managing very large herds can pose logistical challenges in monitoring health, providing resources, or executing rapid movements during crises.

Reviewer #2, Concern #7: Consider comparing your findings with those of similar studies from other countries or regions with comparable pastoral systems. It would be valuable to compare and contrast policy interventions that solely address climatic shocks, as you recommended, with those that combine or bundle climatic and health risk management and report on their various outcomes.

Also, outline a more specific plan for implementing your recommendations, including expected results and potential barriers to implementation.

Authors response to Reviewer #2, Concern #7: We sincerely thank the reviewer for raising this important concern.

In the discussion section we have considered comparing our findings with those of similar studies from other countries or regions with comparable pastoral systems. We have also compared and contrasted policy interventions that solely address climatic shocks, as we recommended, with those that combine or bundle climatic and health risk management and report on their various outcomes. We, also, outlined a more specific plan for implementing your recommendations, including expected results and potential barriers to implementation (See the discussion section).

Reviewer #3

1. Methodology

Reviewer #3, Concern #1: Map of study area could provide valuable context. Some potential information could be reveal; Geographical features. Location of water resources, pastures etc., distribution of camel population and proximity to boarder, roads and other infrastructures. That could help in identify potential relationship between environmental factors and camel targeted interventions.

Authors response to Reviewer #3, Concern #1: We appreciate the reviewer’s insightful suggestion. However, incorporating this map within the current study will be considered in future research to provide deeper spatial insights into the relationship between environmental factors and camel-targeted interventions.

Reviewer #3, Concern #2: Study interaction between variables e.g. includes interaction terms in regression models (Water Source Region) to study the effect of water source effect in region.

Authors response to Reviewer #3, Concern #2: The analysis has been updated to include interaction terms between key predictors (e.g., Water Source × Region). The results and interpretation of these interactions are now presented in the revised regression tables and discussed in the results and discussion sections. This addition helps to capture complex relationships and improves the explanatory power of the model.

2. Data Analysis

Reviewer #3, Concern #3: In Table 1, correct the data order and align gender categories properly.

Authors response to Reviewer #3, Concern #3: The gender of household head variable has been corrected and reordered in Table 1 as suggested. The table now presents the data in the recommended order for clarity

Reviewer #3, Concern #4: in all Tables, Reorder predictors under three main categories Demographic, Socio-economic, and Environmental factors.

Authors response to Reviewer #3, Concern #4: All tables presenting predictors have been reorganized according to the reviewer’s guidance.

Reviewer #3, Concern #5: If data permit, account for seasonal variation in camel mortality rates; otherwise, note it as a recommendation.

Authors response to Reviewer #3, Concern #5: Seasonal variation was explored during data review. Unfortunately, the dataset did not include sufficient temporal details to quantify seasonal changes in mortality. This limitation is now clearly acknowledged in the discussion section, and a recommendation for future studies to include seasonal monitoring has been added to the conclusion.

Reviewer #3, Concern #6: In Table 2, replace “Region 1” with the actual control variable name (e.g., “Awdal”) and ensure consistent formatting.

Authors response to Reviewer #3, Concern #6: The regression tables have been revised accordingly. “Region 1” has been replaced with “Awdal (reference category)” throughout all models (See Table 2).

3. Conclusion

Reviewer #3, Concern #7: In conclusion the study examines each potential determinant as solitary factor without considering potential Interaction between variables. This may overlook complex relationship and synergies between factors, potentially limited the study’s ability to capture the full range of influences.

Authors response to Reviewer #3, Concern #7: This concern has been addressed through the inclusion of interaction terms in the regression models (see Table 3). The revised conclusion now explicitly acknowledges these multidimensional relationships and discusses how socio-demographic and environmental factors jointly contribute to regional differences in camel mortality. The revised text highlights the interlinked influence of water scarcity, pasture degradation, and socio-economic constraints on camel survival.

4. Additional Comment

Reviewer #3, Concern #8: If the study effectively links the socio-demographic and environmental determinants to justify the increase of camel mortality risk in the eastern regions of Somaliland, to explore how these factors interact and impact camel mortality in the regions. it could provide valuable insights for policy makers and stakeholders.

Authors response to Reviewer #3, Concern #8: This has been fully incorporated. The discussion now emphasizes how environmental stressors (such as drought frequency and water scarcity) interact with socio-demographic

---

## [Decision Letter · Decision Letter 1]

12 Nov 2025

PONE-D-25-37649R1Socio-demographic and Environmental Determinants of Camel Mortality in Somaliland's Nomadic Communities: A Count Regression Analysis of the 2020 Demographic and Health SurveyPLOS ONE

Dear Dr. Salih,

Thank you for submitting your manuscript to PLOS ONE. After careful consideration, we feel that it has merit but does not fully meet PLOS ONE’s publication criteria as it currently stands. Therefore, we invite you to submit a revised version of the manuscript that addresses the points raised during the review process.

**ACADEMIC EDITOR:**  Dear Authors, Thank you for your revisions which significantly enhanced the pieces of writing. However, two reviewers have recommended more improvements. Please carefully modify your paper to address all of the comments raised by the reviewers.============================== Please submit your revised manuscript by Dec 27 2025 11:59PM. If you will need more time than this to complete your revisions, please reply to this message or contact the journal office at plosone@plos.org . Please include the following items when submitting your revised manuscript:

We look forward to receiving your revised manuscript.

Kind regards,

Nussieba A. Osman, Dr. Med. Vet.

Academic Editor

PLOS ONE

Journal Requirements:

Reviewers' comments:

Reviewer's Responses to Questions

**Comments to the Author**

1. If the authors have adequately addressed your comments raised in a previous round of review and you feel that this manuscript is now acceptable for publication, you may indicate that here to bypass the “Comments to the Author” section, enter your conflict of interest statement in the “Confidential to Editor” section, and submit your "Accept" recommendation.

Reviewer #1: All comments have been addressed

Reviewer #2: All comments have been addressed

Reviewer #3: (No Response)

2. Is the manuscript technically sound, and do the data support the conclusions?

Reviewer #1: Yes

Reviewer #2: Yes

Reviewer #3: Yes

3. Has the statistical analysis been performed appropriately and rigorously? 

Reviewer #1: Yes

Reviewer #2: Yes

Reviewer #3: Yes

4. Have the authors made all data underlying the findings in their manuscript fully available?

Reviewer #1: Yes

Reviewer #2: Yes

Reviewer #3: No

5. Is the manuscript presented in an intelligible fashion and written in standard English?

Reviewer #1: Yes

Reviewer #2: Yes

Reviewer #3: Yes

6. Review Comments to the Author

Reviewer #1: Good work, kindly check the references and complete the section according to the references cited in the text.

Reviewer #2: Dear Dr. Omran Salih, and Dr. Mohamed Ahmed Hassan,

Thank you for your revisions to the manuscript titled "Socio-demographic and Environmental Determinants of Camel Mortality in Somaliland's Nomadic Communities: A Count Regression Analysis of the 2020 Demographic and Health Survey." The second reviewer appreciates the effort you have put into addressing their comments. Your detailed responses and the incorporation of their suggestions have enhanced the clarity and robustness of the study.

The inclusion of specific diseases affecting camel mortality, the clarification of gender-related findings, and the application of Negative Binomial regression to address over-dispersion are commendable. Additionally, the authors’ explanation of using household water sources and toilet facilities as proxies for environmental variables, although still somewhat “weak,” is well-articulated and provides valuable context for the study.

To further strengthen the manuscript and ensure its readiness for publication, we would like to suggest the following:

1. Add some visual representation of major findings: Consider adding graphs or charts to visually represent the regional disparities in camel mortality and the impact of key determinants such as water sources and household demographics. This will make the findings more accessible to readers.

2. Policy recommendations: While the discussion section provides a good foundation for policy implications, you could expand on specific actionable steps for policymakers. For example, outline how targeted interventions in the eastern regions could be implemented, including pathways such as potential partnerships with local organizations or international agencies.

3. Future research directions: While you have acknowledged the limitations of the dataset, it may be helpful to elaborate on how future studies could address these gaps. For instance, suggest including actual seasonal data, and direct environmental indicators, or longitudinal studies to capture these trends over time.

4. Ethno-veterinary practices: Since traditional knowledge plays a significant role in camel husbandry, you could provide more examples or case studies in the discussion section of successful ethno-veterinary practices that could be integrated with modern interventions.

5. Interactive map: While you have noted that a map will be considered in future research, even a basic map showing the study regions and their environmental features could add valuable context to the current manuscript.

Anyway, the reviewer is confident that these additions will further enhance the manuscript's impact and utility for policymakers, researchers, and stakeholders in veterinary epidemiology and pastoral resilience.

Thank you once again for your dedication to improving the manuscript. We look forward to seeing the final version and the potential contributions your work could make to the field.

Best regards,

Reviewer #2

Reviewer #3: I appreciate the authors’ thoughtful revisions and clear responses. Several earlier concerns have been satisfactorily addressed; however, a number of important points remain that require further clarification or correction:

1. Study-Area Map:

Including a study-area map would significantly enhance the manuscript by illustrating the geographic and environmental context relevant to camel mortality. This would help visualize regional variation, ecological zones, and water sources. Such an addition would not conflict with future spatial analyses but would instead provide essential background for interpreting current results.

2. Although the authors state that interaction terms (e.g., Water Source × Region) were included, these are not reflected in the regression tables or discussed in the results. Consequently, potential multidimensional relationships between predictors remain unexplored, so regional disparities could interpret in terms of environmental or socio-economic context. Which have practical implication; for camel health management or policy; by informing how interventions would be e.g. regional focus or targeted focus improvement of water resources, gender & youth training.

3. Seasonality remains a key factor in camel mortality, as droughts, floods, and disease outbreaks follow known seasonal patterns. Explicit acknowledgment of this aspect in the Discussion section would strengthen interpretation of mortality dynamics, even without detailed temporal data.

Updated Manuscript review

4. Population density in the study area was not considered. Including it or acknowledging its absence as a limitation would help contextualize environmental and management pressures on camel populations.

5. Table 1, titled “Summary Statistics of Key Variables in Camel Mortality Models” appears to serve as a template rather than presenting actual data. The values under “Summary Statistic (Example, e.g.)” seem illustrative, and the “Outcome Variables” section lacks even e.g. statistics, however they are very important. Additionally, related data for other sections appear in Table 2, column 2 (“Total Households n (%)”).

6. The Hygiene Facility variable appears omitted from Table 2, with the group “No Facility” misplaced under Household Wealth Quintile. Please correct.

7. Ensure consistent use of variable names—specifically, harmonize “Type of Toilet Facility” and “Hygiene Facility” across tables and text.

8. Inconsistencies in Regression Tables:

• In the Negative Binomial Regression of Camel Deaths due to Flood (Table 4), the Female coefficient is reported as 0.08 with no significance indicated, these inconsistencies between tables and text should be verified and corrected. In addition, this sentence need to be phrased correctly “Interestingly, being female (coefficient of 1.576, p<0.01) is associated with a significantly higher number of camel death, which might reflect gendered roles in flood response or access to resources.” Also Table 5 (Female Coefficient) do not align with the text.

• In Table 6 The constant term; under Hygiene Facility variable; shows a highly significant and unexpectedly large coefficient (1.338), Please justified

7. PLOS authors have the option to publish the peer review history of their article (what does this mean? ). If published, this will include your full peer review and any attached files.

**Do you want your identity to be public for this peer review?** For information about this choice, including consent withdrawal, please see our Privacy Policy .

Reviewer #1: **Yes:** Asim Faraz

Reviewer #2: No

Reviewer #3: **Yes:** Alia Hassan Mohammed Ahmed

---

## [Author Response · Author response to Decision Letter 2]

9 Dec 2025

Manuscript ID: SO-25-2570

Manuscript Title: Socio-demographic and Environmental Determinants of Camel Mortality in Somaliland's Nomadic Communities: A Count Regression Analysis of the 2020 Demographic and Health Survey

To: PLoS ONE

Re: Response to reviewers

Dear Respected Editor,

Thank you for allowing a resubmission of our manuscript, with an opportunity to address the reviewers’ comments.

We are uploading (a) our point-by-point response to the comments (below) (response to reviewers), (b) an updated manuscript with green highlighting indicating changes, and (c) a clean updated manuscript without highlights.

Finally, we would like to thank the reviewers for reviewing the paper and providing suggestions to improve the paper’s quality. The authors have addressed the comments and suggestions, as described in the response below.

We look forward to your decision.

Best regards,

The paper’s authors.

Reviewer #2:

Reviewer #2, Concern #1: Add some visual representation of major findings: Consider adding graphs or charts to visually represent the regional disparities in camel mortality and the impact of key determinants such as water sources and household demographics. This will make the findings more accessible to readers.

Authors response to Reviewer #2, Concern #1: We appreciate the reviewer’s suggestion. We have now added Figure 1 to visually present the major findings, and we included a dedicated paragraph summarizing these key results within the new section titled Summary of Major Findings (See page 18).

Reviewer #2, Concern #2: Policy recommendations: While the discussion section provides a good foundation for policy implications, you could expand on specific actionable steps for policymakers. For example, outline how targeted interventions in the eastern regions could be implemented, including pathways such as potential partnerships with local organizations or international agencies.

Authors response to Reviewer #2, Concern #2: We thank the reviewer for this valuable comment. In the revised manuscript, we have substantially added specific actionable steps for the policymakers (See page 20).

The findings suggest that infrastructure investments, particularly to support communities transitioning away from unimproved water sources, should be prioritized by the Ministry of Livestock and Rural Development and its partners. In particular, efforts should focus on protecting boreholes and sanitation facilities and mitigating disease vectors in the eastern parts of Sanaag, Sool, and Togdheer. In addition, mobile veterinary clinics and subsidized drug distribution networks should be established with the support of international NGOs and local pastoral associations

Reviewer #2, Concern #3: Future research directions: While you have acknowledged the limitations of the dataset, it may be helpful to elaborate on how future studies could address these gaps. For instance, suggest including actual seasonal data, and direct environmental indicators, or longitudinal studies to capture these trends over time.

Authors response to Reviewer #2, Concern #3: In response, we have expanded the discussion of existing research gaps and outlined how future studies could address these issues (See page 21).

Reviewer #2, Concern #4: Ethno-veterinary practices: Since traditional knowledge plays a significant role in camel husbandry, you could provide more examples or case studies in the discussion section of successful ethno-veterinary practices that could be integrated with modern interventions.

Authors response to Reviewer #2, Concern #4: In response, we have added concrete examples of effective ethno-veterinary practices used by Somali pastoralists. These include the use of cauterization for lameness and chronic conditions, often performed with attention to sterility and pain management to reduce secondary infections, as well as the application of low-cost phytotherapeutic remedies for parasite control and wound healing (see pages 21).

Reviewer #2, Concern #5: Interactive map: While you have noted that a map will be considered in future research, even a basic map showing the study regions and their environmental features could add valuable context to the current manuscript.

Authors response to Reviewer #2, Concern #5: We thank the reviewer for this suggestion. We acknowledge the importance of map. We have developed study area map which is illustrating the geographic context relevant to the camel mortality. See Figure 1. Geospatial analysis of cause-specific. (See Page 19).

Reviewer #3

Reviewer #3, Concern #1: Study-Area Map: Including a study-area map would significantly enhance the manuscript by illustrating the geographic and environmental context relevant to camel mortality. This would help visualize regional variation, ecological zones, and water sources. Such an addition would not conflict with future spatial analyses but would instead provide essential background for interpreting current results.

Authors response to Reviewer #3, Concern #1: We sincerely thank the reviewer for raising this important recommendation. We have developed study area map which is illustrating the geographic context relevant to the camel mortality in Figure 1 (See Page 19).

Reviewer #3, Concern #2: Although the authors state that interaction terms (e.g., Water Source × Region) were included, these are not reflected in the regression tables or discussed in the results. Consequently, potential multidimensional relationships between predictors remain unexplored, so regional disparities could interpret in terms of environmental or socio-economic context. Which have practical implication; for camel health management or policy; by informing how interventions would be e.g. regional focus or targeted focus improvement of water resources, gender & youth training.

Authors response to Reviewer #3, Concern #2: We thank the reviewer for this valuable comment. We agree that the multidimensional relationships between predictors, particularly region and water source are important for fully understanding the disparities in camel mortality. Although interaction terms were initially considered, they were not adequately reflected in the earlier version of the results. To address this concern, we have strengthened the discussion by elaborating on the importance of these potential interactions and how they may shape regional vulnerabilities (see page 21).

Additionally, we have added a regional mortality map (Figure 1) to visually illustrate how mortality rates vary across regions, supporting the argument for context-specific interventions.

The revised discussion now clarifies that environmental and demographic factors were treated as independent contributors to risk, but this additive approach may obscure important interaction effects. For instance, examining the interaction between region and water source could reveal that mortality associated with unimproved water points is disproportionately higher in eastern regions such as Sanaag and Sool. Such patterns are consistent with observations from neighbouring pastoral systems in Ethiopia, where drought-related livestock losses cluster around areas with limited water access. This expanded discussion highlights the practical implications for designing targeted, region-specific livestock health interventions.

Reviewer #3, Concern #3: Seasonality remains a key factor in camel mortality, as droughts, floods, and disease outbreaks follow known seasonal patterns. Explicit acknowledgment of this aspect in the Discussion section would strengthen interpretation of mortality dynamics, even without detailed temporal data.

Authors response to Reviewer #3, Concern #2: We thank the reviewer for this valuable comment. We have explicitly acknowledged the seasonality as a key of camel mortality in the discussion section. See (page 19).

“Moreover, Somaliland’s unique climate further shapes the underlying pattern of risk, prolonged dry season impose severe nutritional constraints, whereas rainy periods facilitate disease spread through herd aggregation and contamination of watersource”.

Reviewer #3, Concern #4: Population density in the study area was not considered. Including it or acknowledging its absence as a limitation would help contextualize environmental and management pressures on camel populations.

Authors response to Reviewer #3, Concern #4: We acknowledged the absence of the population density while we have acknowledged and address it in the study area (See Page 4).

“Somaliland covers an estimated land area of 176,119.2km2 and has a population of about 4.2 million people.”

Reviewer #3, Concern #5: Table 1, titled “Summary Statistics of Key Variables in Camel Mortality Models” appears to serve as a template rather than presenting actual data. The values under “Summary Statistic (Example, e.g.)” seem illustrative, and the “Outcome Variables” section lacks even e.g. statistics, however they are very important. Additionally, related data for other sections appear in Table 2, column 2 (“Total Households n (%)”).

Authors response to Reviewer #3, Concern #5: We thank the reviewer for this valuable recommendation. We have related the data in Table 1, “Summary Statistic (Example, e.g.)” section, with Table 2, column 2 (“Total Households n (%)”). (Page 6)

Reviewer #3, Concern #6: The Hygiene Facility variable appears omitted from Table 2, with the group “No Facility” misplaced under Household Wealth Quintile. Please correct.

Authors response to Reviewer #3, Concern #6: We thank the reviewer for this valuable comments about the corrections. We have corrected the misplaced category, "No Facility," and put it under the Hygiene Facility variable. (Page 9)

Reviewer #3, Concern #7: Ensure consistent use of variable names—specifically, harmonize “Type of Toilet Facility” and “Hygiene Facility” across tables and text.

Authors response to Reviewer #3, Concern #7: We sincerely thank the reviewer for the recommendations. We have harmonized “Type of Toilet Facility” and “Hygiene Facility” across tables and text and use “Hygiene Facility” in the tables and manuscript (Pages 4, 10, 11, 12, 13, 15, 17, 18).

Reviewer #3, Concern #8: Inconsistencies in Regression Tables: In the Negative Binomial Regression of Camel Deaths due to Flood (Table 4), the Female coefficient is reported as 0.08 with no significance indicated, these inconsistencies between tables and text should be verified and corrected. In addition, this sentence need to be phrased correctly “Interestingly, being female (coefficient of 1.576, p<0.01) is associated with a significantly higher number of camel death, which might reflect gendered roles in flood response or access to resources.” Also Table 5 (Female Coefficient) do not align with the text. In Table 6 The constant term; under Hygiene Facility variable; shows a highly significant and unexpectedly large coefficient (1.338), Please justified

Authors response to Reviewer #3, Concern #8: We have corrected the inconsistencies between table 4 and its text. We have also aligned the Table 5 (Female Coefficient) with its text. For justifying, the constant term under the Hygiene facility variable is the regression intercept. It represents the expected log-count of the dependent variable “Camel Death” when all independent variables are set to their references categories. (Pages 12, 14 and 17).

---

## [Decision Letter · Decision Letter 2]

5 Jan 2026

PONE-D-25-37649R2Socio-demographic and Environmental Determinants of Camel Mortality in Somaliland's Nomadic Communities: A Negative Binomial Regression Analysis of the 2020 Demographic and Health SurveyPLOS One

Dear Dr. Salih,

Thank you for submitting your manuscript to PLOS ONE. After careful consideration, we feel that it has merit but does not fully meet PLOS ONE’s publication criteria as it currently stands. Therefore, we invite you to submit a revised version of the manuscript that addresses the points raised during the review process.

**ACADEMIC EDITOR:**  Dear Authors, Thank you for revising your manuscript and addressing the majority of the reviewers' comments. However, minor revisions are required for improving your manuscript. Please carefully revise the manuscript with regard to all comments raised by the reviewers. Please submit your revised manuscript by Feb 19 2026 11:59PM. If you will need more time than this to complete your revisions, please reply to this message or contact the journal office at plosone@plos.org . Please include the following items when submitting your revised manuscript:

We look forward to receiving your revised manuscript.

Kind regards,

Nussieba A. Osman, Dr. Med. Vet.

Academic Editor

PLOS One

Journal Requirements:

Reviewers' comments:

Reviewer's Responses to Questions

**Comments to the Author**

1. If the authors have adequately addressed your comments raised in a previous round of review and you feel that this manuscript is now acceptable for publication, you may indicate that here to bypass the “Comments to the Author” section, enter your conflict of interest statement in the “Confidential to Editor” section, and submit your "Accept" recommendation.

Reviewer #2: All comments have been addressed

Reviewer #3: All comments have been addressed

2. Is the manuscript technically sound, and do the data support the conclusions?

Reviewer #2: Yes

Reviewer #3: Yes

3. Has the statistical analysis been performed appropriately and rigorously? 

Reviewer #2: Yes

Reviewer #3: Yes

4. Have the authors made all data underlying the findings in their manuscript fully available?

Reviewer #2: (No Response)

Reviewer #3: Yes

5. Is the manuscript presented in an intelligible fashion and written in standard English?

Reviewer #2: Yes

Reviewer #3: Yes

6. Review Comments to the Author

Reviewer #2: Dear Authors,

Happy New Year 2026!

Thank you for the opportunity to review again your manuscript titled "Socio-demographic and Environmental Determinants of Camel Mortality in Somaliland's Nomadic Communities: A Negative Binomial Regression Analysis of the 2020 Demographic and Health Survey." Your updated work provides a valuable contribution to understanding how social, demographic, and environmental factors influence camel mortality in the Somaliland and pastoral communities. The use of nationally representative data with now the application of Negative Binomial Regression models are commendable and well-suited to the study's objectives.

Reviewer 2 appreciates your efforts in addressing their previous comments and suggestions, especially those related to the use of visuals, including graphs and maps, to showcase some key geographical results more effectively and better convey the insights generated.

My only comment at this stage is to standardize the presentation of the presentations of the tables. In Table 1, there are multiple rows of the outcome variables with missing values of Mean (SD), Min, Max, and Median (IQR). Additionally, there are still some green highlights in the cleaned version of the paper; kindly proofread the manuscript one more time to remove them and correct typos and grammatical errors.

I wish you the best of luck and well done again!

Best,

Reviewer2

Reviewer #3: The paper addresses a highly relevant topic and has strong potential to inform livestock management and policy decisions in the studied regions. The analysis is generally sound, and the manuscript has improved substantially through the revision process. I support its acceptance for publication.

I have only one minor point for clarification in Table 1. The outcome variables (e.g., Total Camel Deaths and Camel Deaths due to Drought, Flood, or Disease) are listed as count variables, but no information is provided regarding the reference period or unit of measurement (e.g., deaths per household, per year, or per herd size). For clarity and ease of interpretation, the authors may either (i) briefly specify this information in the table or accompanying note, or (ii) omit this part of the table if the outcome variables are not intended to be summarized descriptively.

7. PLOS authors have the option to publish the peer review history of their article (what does this mean? ). If published, this will include your full peer review and any attached files.

**Do you want your identity to be public for this peer review?** For information about this choice, including consent withdrawal, please see our Privacy Policy .

Reviewer #2: No

Reviewer #3: **Yes:** Alia Hassan Mohammed Ahmed

---

## [Author Response · Author response to Decision Letter 3]

11 Jan 2026

Manuscript ID: SO-25-2570

Manuscript Title: Socio-demographic and Environmental Determinants of Camel Mortality in Somaliland's Nomadic Communities: A Negative Binomial Regression Analysis of the 2020 Demographic and Health Survey

To: PLoS ONE

Re: Response to reviewers

Dear Respected Editor,

Thank you for allowing a resubmission of our manuscript, with an opportunity to address the reviewers’ comments.

We are uploading (a) our point-by-point response to the comments (below) (response to reviewers), (b) an updated manuscript with yellow highlighting indicating changes, and (c) a clean updated manuscript without highlights.

Finally, we would like to thank the reviewers for reviewing the paper and providing suggestions to improve the paper’s quality. The authors have addressed the comments and suggestions, as described in the response below.

We look forward to your decision.

Best regards,

The paper’s authors.

Reviewer #2

Reviewer #2, Concern #1: My only comment at this stage is to standardize the presentation of the presentations of the tables. In Table 1, there are multiple rows of the outcome variables with missing values of Mean (SD), Min, Max, and Median (IQR). Additionally, there are still some green highlights in the cleaned version of the paper; kindly proofread the manuscript one more time to remove them and correct typos and grammatical errors.

Authors response to Reviewer #2, Concern #1: We thank the reviewer for this valuable comment. We have omitted this part of the table as suggested by reviewer #3, and to avoid repetition. We have also proofread the manuscript and removed the highlights.

Reviewer #3

Reviewer #3, Concern #1: I have only one minor point for clarification in Table 1. The outcome variables (e.g., Total Camel Deaths and Camel Deaths due to Drought, Flood, or Disease) are listed as count variables, but no information is provided regarding the reference period or unit of measurement (e.g., deaths per household, per year, or per herd size). For clarity and ease of interpretation, the authors may either (i) briefly specify this information in the table or accompanying note, or (ii) omit this part of the table if the outcome variables are not intended to be summarized descriptively.

Authors response to Reviewer #3, Concern #1: We sincerely thank the reviewer for raising this important point. We have chosen the option you have given us, which says, “Omit this part of the table if the outcome variables are not intended to be summarized descriptively.” And we omit that part.

---

## [Editor Report · Decision Letter 3]

20 Jan 2026

Socio-demographic and Environmental Determinants of Camel Mortality in Somaliland's Nomadic Communities: A Negative Binomial Regression Analysis of the 2020 Demographic and Health Survey

PONE-D-25-37649R3

Dear Dr. Salih,

We’re pleased to inform you that your manuscript has been judged scientifically suitable for publication and will be formally accepted for publication once it meets all outstanding technical requirements.

Kind regards,

Nussieba A. Osman, Dr. Med. Vet.

Academic Editor

PLOS One

Additional Editor Comments (optional):

Dear Authors,

congratulation on accepting your manuscript for publications. I would like to raiser few minor comments from the reviewers to revise tables.

Congratulations on accepting your manuscript for publication. I would like to address a few minor comments from the reviewers regarding the tables.
---

## [Editor Report · Acceptance letter]

PONE-D-25-37649R3

PLOS One

Dear Dr. Salih,

I'm pleased to inform you that your manuscript has been deemed suitable for publication in PLOS One. Congratulations! Your manuscript is now being handed over to our production team.

Kind regards,

on behalf of

Dr. Nussieba A. Osman

Academic Editor

PLOS One